# Machine Learning and Artificial Intelligence in Intensive Care Medicine: Critical Recalibrations from Rule-Based Systems to Frontier Models

**DOI:** 10.3390/jcm14124026

**Published:** 2025-06-06

**Authors:** Pierre Hadweh, Alexandre Niset, Michele Salvagno, Mejdeddine Al Barajraji, Salim El Hadwe, Fabio Silvio Taccone, Sami Barrit

**Affiliations:** 1Sciense, New York, NY 10027, USA; pierrehadweh@gmail.com (P.H.); mejdi.albarajraji@gmail.com (M.A.B.); salimhadweh15@gmail.com (S.E.H.); 2Pediatric Intensive Care Unit, Cliniques Universitaires Saint-Luc, 1200 Bruxelles, Belgium; 3Department of Intensive Care, Hôpital Universitaire de Bruxelles (HUB), Université Libre de Bruxelles (ULB), Anderlecht, 1070 Brussels, Belgium; michele.salvagno1@gmail.com (M.S.); fabio.taccone@ulb.be (F.S.T.); 4Department of Neurosurgery, CHR Citadelle, 4000 Liège, Belgium; 5Department of Clinical Neurosciences, University of Cambridge, Cambridge CB2 1TN, UK; 6Bioelectronics Laboratory, Department of Electrical Engineering, University of Cambridge, Cambridge CB2 1TN, UK; 7Institut Neuroscience des Systèmes (INS), Aix-Marseille Université, 13005 Marseille, France

**Keywords:** intensive care medicine, critical care, clinical decision support systems, artificial intelligence, machine learning, sepsis, predictive analytics, deep learning, natural language processing, safety, privacy, large language models

## Abstract

Artificial intelligence (AI) and machine learning (ML) are rapidly transforming clinical decision support systems (CDSSs) in intensive care units (ICUs), where vast amounts of real-time data present both an opportunity and a challenge for timely clinical decision-making. Here, we trace the evolution of machine intelligence in critical care. This technology has been applied across key ICU domains such as early warning systems, sepsis management, mechanical ventilation, and diagnostic support. We highlight a transition from rule-based systems to more sophisticated machine learning approaches, including emerging frontier models. While these tools demonstrate strong potential to improve predictive performance and workflow efficiency, their implementation remains constrained by concerns around transparency, workflow integration, bias, and regulatory challenges. Ensuring the safe, effective, and ethical use of AI in intensive care will depend on validated, human-centered systems supported by transdisciplinary collaboration, technological literacy, prospective evaluation, and continuous monitoring.

## 1. Introduction

The intensive care unit (ICU) represents both an ideal testing ground and a significant challenge for artificial intelligence (AI) applications in medicine. ICUs generate vast quantities of high-dimensional data, from continuous physiological monitoring, frequent laboratory testing, imaging studies, and detailed clinical documentation, which creates both an opportunity and a necessity for advanced computational approaches to data analysis and decision support [1]. The complexity of critically ill patients, characterized by multisystem disease, rapid clinical changes, and high-stakes decision-making under significant time constraints, necessitates the expertise of highly specialized clinicians who must continuously update their knowledge and skills to effectively integrate evolving technologies and maintain optimal patient care standards [2].

Clinical decision support systems (CDSSs) aim to augment human capabilities by providing actionable insights at the point of care [3]. While traditional CDSSs have often relied on rule-based logic derived from established guidelines or expert consensus, the advent of machine learning (ML) techniques has enabled a paradigm shift toward data-driven approaches capable of detecting complex patterns and subtle signals that might elude human perception [4,5]. This evolution in CDSS technology aligns with the growing recognition that conventional scoring systems in critical care have limitations in capturing the dynamic nature of critical illness and personalizing risk assessment [6,7].

Although AI tools have shown promise in many areas of medicine, their use in ICU settings remains limited and inconsistent. Key barriers include a lack of model transparency, unclear regulatory pathways, and limited clinician trust [8]. These challenges underscore the urgent need for stronger clinical evidence, rigorous validation studies, and ethical design principles to guide the safe and effective integration of AI into critical care practice [9].

This review examines the integration of machine intelligence in intensive care medicine, from early rule-based systems to current applications leveraging advanced machine learning and natural language processing for AI-driven CDSS. We evaluate the evidence supporting various AI/ML modalities across specific domains of critical care practice, including early warning systems, sepsis prediction and management, mechanical ventilation optimization, and clinical documentation. Implementation challenges, spanning technical integration, clinical workflow adaptation, ethical considerations, and regulatory frameworks, are critically analyzed. Finally, we discuss future directions that may influence the trajectory of AI in critical care, stressing the need for transparent, clinically validated systems that complement rather than supplant clinical judgment.

## 2. Historical Evolution of AI in Critical Care

Machine intelligence in biomedicine emerged during the 1960s with seminal efforts to develop expert systems emulating human expert decision-making. In the 1970s, such systems were applied in critical care with initiatives such as MYCIN, developed to guide antimicrobial therapy for severe infections (Figure 1 and Table 1) [10]. These systems encoded medical knowledge through explicitly programmed IF–THEN rules, allowing the computer to mimic clinical reasoning within narrow domains. Despite showing promise in research settings, these early systems faced practical limitations in handling the complexity and uncertainty inherent in critical care, and few achieved widespread clinical implementation [11].

The transition from rule-based to data-driven approaches began in the 1990s, with the development of ICU-specific prognostic models using statistical methods, such as APACHE (Acute Physiology and Chronic Health Evaluation) and SAPS (Simplified Acute Physiology Score) [34]. While groundbreaking, these models relied on relatively simple logistic regression techniques and required periodic manual recalibration to maintain accuracy across changing patient populations [24].

The modern era of AI in critical care emerged in the 2010s with the convergence of several key factors: (I) widespread adoption of electronic health records (EHRs) creating large, structured clinical datasets; (II) advances in computational power enabling complex model training; (III) development of more sophisticated machine learning algorithms capable of handling high-dimensional data with non-linear relationships; and (IV) the creation of publicly available critical care databases such as the Medical Information Mart for Intensive Care (MIMIC), which fostered open research and benchmarking [35,36].

## 3. AI/ML Paradigms and Technologies

Contemporary AI in ICU employs distinct machine learning paradigms, each aligned with specific clinical changes (Figure 2 and Table 2).

### 3.1. Supervised Learning

Supervised learning models use labeled patient data that include patient features (such as vital signs or lab test results) along with known clinical outcomes to train predictive models [42]. It is the most common AI method in critical care. This approach includes many ICU decision-support tools that learn from large datasets of ICU patient records. These tools can estimate a patient’s mortality risk, predict organ failure, or generate early warnings of deterioration [43]. Modern supervised methods, such as ensemble classifiers and deep neural networks, often outperform traditional scoring systems in capturing complex nonlinear relationships in the data. For example, a supervised sepsis early-warning system that can alert clinicians hours before a patient meets clinical sepsis criteria, effectively providing a critical lead-time for intervention [44].

That said, supervised models are only as good as their training data. They require extensive, high-quality labeled datasets and can inherit biases present in those data. If certain patient groups or outcomes are underrepresented or recorded with bias, the model’s predictions may be skewed or less generalizable [45,46]. Thus, careful dataset curation, bias correction, and continuous performance monitoring are essential when deploying supervised learning in intensive care. These measures help ensure that the model’s recommendations remain accurate and fair across diverse patient populations and clinical scenarios.

### 3.2. Unsupervised Learning

Unsupervised learning methodologies focus on finding hidden structure in unlabelled datasets to identify latent patterns, offering valuable insights by uncovering novel patient phenotypes. The goal is to let the data themselves reveal natural clusters of patients or important variables, rather than relying on prior assumptions. This approach is especially valuable for complex conditions such as sepsis and acute respiratory distress syndrome (ARDS) [29,30].

Key unsupervised techniques in critical care include the following:Clustering algorithms (e.g., *k*-means, hierarchical clustering): These group patients with similar clinical profiles or characteristics.Dimensionality reduction techniques (e.g., principal component analysis (PCA) or deep autoencoders), which compress high-dimensional data (such as hundreds of lab values and vital signs) into a smaller set of informative features, reducing complexity while preserving important information.

Recent research has applied unsupervised learning methods to uncover meaningful patient subgroups within ICU populations, including the following:Sepsis Phenotypes: Using clustering analyses on sepsis patient data, researchers have uncovered distinct subgroups of sepsis patients that were not previously recognized. These data-driven clusters differ significantly in their clinical characteristics and outcomes [29].ARDS Subphenotypes: unsupervised analyses have identified at least two consistent subphenotypes, often described informally as “hyperinflammatory” and “hypoinflammatory”. Each of these subgroups has its own pattern of biomarker levels, its own severity of illness, and a different response to treatments [38].

In summary, unsupervised learning in intensive care is a powerful tool for knowledge discovery. By uncovering latent patterns in complex ICU data, it can generate new hypotheses, such as novel disease subtypes, enabling more personalized approaches to patient management and deepening our understanding of complex disease beyond what traditional analyses can achieve [29,30,38].

### 3.3. Reinforcement Learning

Reinforcement learning (RL) is an approach to AI where an agent learns by interacting with its environment. The agent makes a sequence of decisions and improves its strategy based on feedback (rewards or penalties). Unlike supervised learning that relies on fixed labeled examples, RL continuously refines its decisions through trial and error. This approach is particularly well-suited to the ICU setting, where clinicians must continually adjust treatments as patient conditions change [47,48].

RL algorithms have been applied to tasks like drug titration and ventilator management. The goal in these applications is to discover an optimal treatment strategy that maximizes patient benefit throughout an ICU stay [49,50].

Classical RL algorithms such as Q-learning estimate the value of each action in a given clinical state, for example, evaluating the long-term benefit of giving an extra dose of a vasopressor to a hypotensive patient. More advanced deep RL methods (such as Deep Deterministic Policy Gradient (DDPG) or Proximal Policy Optimization (PPO)) use neural networks to handle complex, continuous action spaces (for example, continuously adjusting an infusion rate), effectively approximating the optimal policy [49,50]. In recent studies, researchers applied RL to mechanical ventilation by training an agent to adjust ventilator settings. The agent learned to prioritize lung-protection strategies more consistently than standard protocols. In sedation management, researchers developed an RL algorithm to dynamically recommend doses of the sedative dexmedetomidine, aiming to minimize delirium risk. The “AI Clinician” exemplifies this approach, utilizing RL to derive optimal fluid and vasopressor strategies from retrospective ICU data, demonstrating promise in managing pathological conditions in which fluid therapy is relevant, such as sepsis [27].

Practical translation to bedside implementation remains constrained by computational complexity, safety concerns, and integration challenges [39]. Current research increasingly focuses on Human-in-the-Loop RL, whereby clinicians interact with the learning system during training or inference phases. This approach enables models to incorporate expert feedback, enhance interpretability, and ensure safety in high-stakes environments, such as critical care [51,52].

### 3.4. Data Infrastructure and Computational Frameworks

The implementation of these AI paradigms in clinical practice relies on robust data infrastructure and computational frameworks. Comprehensive data repositories, including the MIMIC databases, such as the MIMIC-II and MIMIC-III databases, have been instrumental in advancing ICU AI research by providing extensive, detailed patient records that support robust model training and validation [23]. These repositories contain rich, multimodal data that capture the complexity of critical care, including physiological time series, laboratory results, medication administrations, and clinical documentation. Building upon these data foundations, various machine learning frameworks have been used to address the unique challenges of critical care data. Traditional algorithms, such as random forests and gradient boosting machines, effectively handle structured clinical data, while deep learning architectures—particularly convolutional neural networks (CNNs) and recurrent neural networks (RNNs)—excel at analyzing high-dimensional and temporal ICU data. These advanced architectures have demonstrated promising performance in modelling the complex temporal dynamics characteristic of critical illness progression [23,26,53]. As the field matures, privacy-preserving methods, such as federated learning, have emerged to address the ethical and regulatory challenges associated with healthcare data utilization. These approaches enable collaborative model training across multiple institutions while maintaining patient confidentiality, facilitating the development of generalized AI models across diverse populations without requiring centralized data aggregation [12]. This technological advancement represents a crucial step toward responsible AI deployment in healthcare settings.

Despite the strengths of open ICU datasets and federated learning, significant data infrastructure challenges continue to limit the real-time deployment of AI in clinical settings. Public ICU datasets (such as MIMIC) are constrained by their single-center scope and outdated information. For instance, MIMIC-IV data ends in 2019, before the COVID-19 pandemic [45,54]. In practice, many hospitals lack robust pipelines for streaming high-frequency ICU data (e.g., continuous vital signs and waveforms) into analytics engines. This issue is further compounded by limited on-site computational resources, such as the absence of GPU servers and unreliable network connectivity, which restrict the deployment of complex models at the bedside [55].

## 4. AI Applications in ICUs: Implementation and Challenges

AI technologies are increasingly being deployed across various ICU domains, from early warning systems and sepsis prediction to ventilator management and diagnostic support. These applications demonstrate the growing impact of machine intelligence in frontline care, each with distinct implementation approaches and challenges. Table 3 summarizes key performance benchmarks, documented benefits, and operational considerations across these domains.

### 4.1. Early Warning Systems

Early detection of clinical deterioration represents one of the most mature applications of AI in critical care. Traditional early warning score systems, such as the Modified Early Warning Score (MEWS) and the National Early Warning Score (NEWS), rely on simple aggregation of abnormal vital signs. However, these traditional scores reflect the patient’s current risk, without accounting for potential risk trajectories. By contrast, AI-based approaches can incorporate complex interactions between variables, temporal trends, and subtle patterns preceding acute events, on a beat-to-beat basis [57]. Deep learning early warning systems show a substantially higher sensitivity than conventional methods at equivalent specificity, with some studies reporting up to a 250% relative increase [56].

Numerous ML-based early warning systems have demonstrated superior performance compared to traditional scoring methods, with few exceptions [68]. For example, the Electronic Cardiac Arrest Risk Triage (eCART) system was developed using gradient-boosted machines on data from more than 250,000 hospitalizations, significantly outperformed MEWS in predicting cardiac arrest and ICU transfer (area under the curve, AUC of 0.85 vs. 0.70) [58]. When implemented as part of a rapid response team activation system, eCART was associated with a 16% reduction in cardiac arrests [37].

More sophisticated approaches using deep learning on continuous monitoring data have pushed prediction horizons further. Hyland et al. developed a recurrent neural network model analyzing minute-by-minute vital signs that predicted clinical interventions with an AUC of 0.90 up to 6 h before the event [31]. These systems increasingly incorporate strategies to balance sensitivity with specificity and to provide explanations for their predictions to support clinical adoption [69].

Beyond ICU-specific models, recent work in general medical risk prediction has introduced methodological innovations that could be adapted to critical care. For example, HeartEnsembleNet integrates multiple base classifiers with the SHapley Additive exPlanations (SHAP)-based interpretability, achieving state-of-the-art performance for cardiovascular-risk assessment while retaining model explainability [70]. Similarly, a multi-layer perceptron trained with Adadelta, RMSProp, and AdaMax optimizers reached approximately 95% accuracy in early stroke detection tasks [71]. While these systems were developed outside ICU settings, they illustrate transferable strategies that may further enhance early-warning performance when tailored to the dynamic and high-noise environment of the ICU.

### 4.2. Sepsis Care

Sepsis represents an important target for AI applications, given its prevalence, mortality impact, and the established benefit of early intervention. Several machine learning models have been developed for early sepsis detection, with reported promising lead times of 4–12 h before clinical recognition [72].

The Targeted Real-time Early Warning System (TREWS) exemplifies the potential of AI for sepsis care. This random forest-based model uses EHR data to identify patients at risk of sepsis who have undergone rigorous prospective evaluation [73]. In a multi-site study involving over 590,000 patient encounters, Adams et al. demonstrated that when clinicians acted on TREWS alerts within 3 h, in-hospital mortality was reduced by 3.3% absolute percentage points compared to unaddressed alerts [32]. This study provides rare prospective evidence that an AI-driven alert, when integrated into clinical workflow with appropriate response protocols, can meaningfully impact patients’ outcomes.

However, implementation experience has been mixed. The widely deployed Epic Sepsis Model (ESM) has faced criticism after independent validation revealed poor performance (sensitivity 33%, positive predictive value 12%) [59]. This illustrates the importance of transparent, peer-reviewed validation before the widespread implementation of AI tools in critical care.

Beyond detection, AI approaches to sepsis management have also emerged. The “AI Clinician” by Komorowski et al. used reinforcement learning to suggest individualized fluid and vasopressor dosing strategies [27]. Retrospective analysis suggested that when actual clinical decisions matched the AI’s recommendations, mortality was lower. While interesting, these findings remain to be validated in prospective interventional studies.

The TREWS sepsis alert system at Johns Hopkins continuously monitors vital signs, laboratory results, and clinical notes to detect sepsis. Compared with conventional methods, TREWS detected sepsis 6 h earlier and reduced time to first antibiotic by 1.85 h (median). In a two-year study of over 500,000 patient encounters, this was associated with a 20% relative reduction in sepsis mortality. The system provided alert explanations and adapted to unit workflows, leading to adoption by thousands of providers in routine care [32,74,75].

### 4.3. Ventilation Management

Mechanical ventilation represents another promising domain for AI applications, given the complexity of ventilator management and the potential impact of optimized settings on patient outcomes. AI-guided systems, including fully automated closed-loop devices and decision support tools, optimize mechanical ventilation by adjusting key respiratory parameters and reducing intervention frequency.

Reinforcement learning algorithms have been developed to recommend personalized ventilator settings based on patient characteristics and physiological responses. Peine et al. demonstrated that a reinforcement learning-based approach could recommend positive end-expiratory pressure (PEEP) levels that achieved better oxygenation, while minimizing harmful high plateau pressures compared to standard care in a retrospective analysis [39]. Similar models have been developed for weaning protocols, where ML algorithms incorporate multiple parameters to predict successful extubation with greater accuracy than traditional criteria [76].

Deep learning models analyzing ventilator waveforms have demonstrated high accuracy (>90%) in automatically detecting various forms of asynchrony, potentially enabling real-time monitoring and adjustment [61]. Gholami et al. used random forest classifiers to identify patient–ventilator cycling asynchrony with accuracy comparable to expert clinicians but offering continuous monitoring capability [60].

AI systems have also been implemented to monitor adherence to lung-protective ventilation strategies and suggest adjustments when patients deviate from recommended parameters. For instance, fully automated ventilator systems may reduce driving pressure and mechanical power while improving lung compliance [77]. These systems analyze not just current settings but also trends in respiratory mechanics to customize recommendations based on individual patient characteristics [78].

While most of these applications remain in research settings or limited deployments, full closed-loop systems face both technical and regulatory hurdles, making human-in-the-loop approaches more likely in the near term [79].

At Chi Mei Medical Center in Tainan City, Taiwan, an AI decision support system assisted ICU mechanical ventilation weaning using a two-stage machine learning model to identify optimal extubation timing. Patients managed with AI guidance spent 21 fewer hours on mechanical ventilation compared to standard care. The bedside dashboard tool achieved high staff acceptance, with respiratory therapists and intensivists reporting confidence in the AI’s weaning recommendations. This approach reduced ventilator-associated complications and eased ICU resource utilization. Initial clinician reluctance was addressed through transparent model design and frontline staff involvement in implementation [80].

### 4.4. Diagnostic Support

Machine learning algorithms yield promising diagnostic performance for complex cases and for interpreting diagnostic results. Deep learning models have reached expert-level performance in interpreting standard ICU imaging studies. For chest radiographs, CNNs can detect pneumothorax, misplaced tubes and lines, pulmonary oedema, and consolidation, with high accuracy [40]. These tools can prioritize critical findings for radiologist review or provide immediate feedback when specialist interpretation is unavailable.

Beyond standard vital sign monitoring, AI systems can extract additional diagnostic information from continuous waveforms. Advanced analysis of electrocardiograms can detect subtle signs of ischemia or electrolyte abnormalities, while arterial waveform analysis can provide insights into fluid responsiveness and cardiac function [81].

Large language models (LLMs) and knowledge-based systems can generate differential diagnoses based on clinical data, potentially identifying rare conditions or unusual presentations. Chen et al. [82] demonstrated that an LLM fine-tuned on clinical cases could generate appropriate differential diagnoses for complex ICU scenarios with performance comparable to junior critical care physicians. Similarly, in a study by McDuff et al., a large language model optimized for diagnostic reasoning outperformed unassisted clinicians in standalone performance (top 10 accuracy of 59.1% vs. 33.6%) [62].

ML models can identify patterns in laboratory data that suggest specific diagnoses or predict complications. For example, unsupervised learning approaches have been used to identify distinct metabolic phenotypes in critically ill patients that correlate with outcomes and responses to specific interventions [83].

The integration of these capabilities into coherent diagnostic support systems remains an active area of development, with emphasis on explanatory features that allow clinicians to understand the rationale behind AI suggestions.

### 4.5. Documentation and Workflow

Administrative and documentation burdens represent significant challenges in modern ICU practice. AI applications in this domain aim to reduce cognitive load and time waste, allowing clinicians to focus more on direct patient care.

Natural language processing (NLP) technologies have revolutionized the analysis of unstructured clinical documentation. Transformer-based models such as Bidirectional Encoder Representations from Transformers (BERT) efficiently extract actionable clinical information from extensive physician and nursing notes, converting narrative text into structured, analyzable data [33,83]. This application significantly enhances clinical decision support by incorporating previously inaccessible information embedded in clinical narratives.

NLP systems can generate structured documentation from clinician–patient interactions, team discussions, or even ambient conversations during rounds [84]. Furthermore, automatic speech recognition technologies could reduce documentation burdens by accurately transcribing clinical dictations. Transformer-based models, such as ConstDecoder, enhance transcription quality in acoustically challenging ICU environments, demonstrating practical utility and substantial reduction in documentation errors [64].

AI-powered search and summarizing tools can rapidly retrieve relevant information from voluminous patient records [85]. These can be useful both for research purposes and for in-depth clinical evaluations of patients who have had prolonged hospital stays.

Machine learning models can analyze patterns of ICU workflow to identify inefficiencies and suggest process improvements. AI-based predictive models for ICU length of stay and discharge readiness may improve resource utilization and patient flow [66]. Overall, specific efficiency improvements include a 15% increase in data analysis speed, a 10% faster alert generation, and a 5% improvement in decision-making speed [67].

These applications encounter fewer regulatory barriers compared to those designed for direct clinical decision support, and they may serve as a gateway for AI adoption in ICUs.

### 4.6. Clinical Decision Support

AI-CDSS aims to guide complex interventions, including antibiotic prescription and fluid management. AI-supported antimicrobial stewardship tools exemplify successful clinical implementation with demonstrable positive impact on patient outcomes [13].

Prognostic scoring and risk stratification systems incorporate AI models into ICU workflows, surpassing traditional scoring systems in both accuracy and adaptability [2,3,24].

These AI-driven tools continuously refine prognostic assessments based on real-time patient data, enabling dynamic risk stratification that adapts to evolving clinical scenarios [6,7,18,24].

Clinical evaluations at Our Lady of the Lake Regional Medical Center implemented IntelliSep, an FDA-cleared AI diagnostic tool for sepsis management. The system analyzes immune cell activation from a single blood sample, generating risk stratification scores within minutes of ICU or emergency department admission. Implementation was associated with a 20% reduction in sepsis mortality and a nearly 2-day decrease in average ICU length of stay for septic patients. Over six months, the hospital performed 1800 fewer blood cultures and reduced nursing labor by nine full days. The tool was integrated into existing clinical dashboards, with staff training on score interpretation [86].

Beyond sepsis management, decision support systems extend to other critical domains, including hemodynamic management, glucose control, and transfusion decisions. These systems aim to implement evidence-based practices while accounting for individual patient variations and contextual factors that might warrant deviation from standard protocols.

### 4.7. Operational Efficiency

At the operational level, resource optimization and workflow management applications employ AI to predict patient acuity and ICU demands, optimizing resource allocation, including bed utilization, staffing, and patient triage, thereby enhancing operational efficiency across critical care units [61].

These systems demonstrate how AI can simultaneously improve clinical outcomes and operational efficiency, addressing the dual challenges of clinical complexity and resource constraints in modern healthcare systems. By predicting patient flow, length of stay, and required resources, these tools help administrators make data-driven decisions about staffing levels, bed allocations, and patient transfers.

Computer vision (CV) technologies enable continuous patient monitoring via non-invasive sensors [87,88]. Advanced algorithms, such as YOLOv8, quantify patient mobility and room activity, enabling improved prediction of clinical events, including delirium and physiological instability, thereby reducing clinician workload [67].

### 4.8. Frontier Models

The latest evolution in AI technology relevant to critical care involves foundation models, large-scale neural networks pre-trained on massive datasets that can be fine-tuned for specific downstream tasks. LLMs and clinical adaptations, such as the Medical Pathways Language Model (Med-PaLM), represent this new paradigm [41]. These models have demonstrated remarkable capabilities in medical knowledge representation, clinical reasoning, and natural language understanding that could transform multiple aspects of critical care decision support [89].

Foundation models offer several potential advantages for critical care applications, including (I) transfer learning from general clinical knowledge to specific ICU tasks with relatively small fine-tuning datasets; (II) multimodal integration of text, tabular data, and potentially imaging; (III) more natural interaction through conversational interfaces; and (IV) sophisticated contextual understanding of complex clinical scenarios [90]. However, they also present unique challenges, including concerns about hallucination (generating plausible but factually incorrect information), transparency limitations, and difficulties in real-time deployment within existing EHR infrastructures [91].

### 4.9. Implementation Challenges

Implementing AI in intensive care units presents complex challenges that span technical, clinical, ethical, and regulatory domains (Table 4). The technical integration of AI systems is complicated by the fragmented nature of ICU infrastructure, where multiple monitoring and information systems often operate in isolation with limited interoperability [92]. This creates substantial barriers to assembling the high-quality, synchronized data streams necessary for robust AI model development and deployment.

As machine learning requires large volumes of high-quality training data, it is essential to ensure that datasets are representative of the target patient populations. Healthcare environments can contain various types of bias and noise that can cause models trained in one hospital setting to perform poorly elsewhere. A prominent concern is alert fatigue, as ICU clinicians already contend with numerous alerts from monitoring systems. False alarms, often caused by missing values due to irregular laboratory sampling, device disconnections, along with artifacts during continuous monitoring, can further mislead algorithms and contribute to work overload [85,96]. The temporal alignment of diverse data elements presents additional complexities.

Effective AI decision support depends on both timely recommendations and user-friendly interface design. Because clinical decisions often occur under intense time pressure, AI systems must provide recommendations at the right moment and present information in a way that is both comprehensive and cognitively accessible. Achieving this requires a nuanced understanding of clinical processes, decision points, and user interface design tailored to urgent care settings [97,98].

Many ML models are considered “black boxes”, generating predictions without offering clear explanations. This lack of transparency poses a significant challenge in critical care settings, where clinicians must take full responsibility for decisions that directly affect patient outcomes [99]. Researchers have explored methods to explain ML models, including SHAP and LIME, which generate interpretations of model behavior [100]; however, these explanations are often unreliable or even misleading. Others advocate for inherently interpretable models, though these may sacrifice some predictive accuracy and require considerable effort to build [101].

Contrastive explanations represent another promising approach, explaining why an algorithm produces one output rather than another [102]. Knowledge-augmented AI systems that combine data-driven learning with explicit medical knowledge graphs or causal models can provide explanations that align with clinical reasoning patterns [103].

Critical care AI applications are subject to heightened ethical and regulatory scrutiny due to their potential impact on vulnerable patients. A primary consideration concerns fairness and bias, as AI systems trained on historical data may perpetuate or amplify existing disparities in critical care [104].

While AI models have shown impressive performance, most applications rely on retrospective data collected for research purposes. Validating their real-world utility requires prospective clinical trials to assess model reliability in real-world heterogeneous and noisy clinical environments. Bridging the gap between AI research and clinical practice will also require multi-center validation, improved model explainability, and workflow-centered implementation strategies to ensure meaningful bedside benefits [105,106].

Regulatory challenges persist as machine learning models evolve rapidly through the incorporation of new data [107]. The FDA’s regulatory framework for AI/ML-based Software considers both intended use and associated risk level. Liability questions remain when AI recommendations contribute to adverse outcomes, with clinicians currently retaining ultimate decision responsibility [108].

Establishing secure systems for collecting, storing, and sharing EHRs is complex. Privacy-preserving methods, such as third-party cloud services, can help safeguard patient information. Federated learning offers a promising solution by enabling decentralized AI training across institutions or devices connected through the Internet of Things (IoT) without transferring raw patient data, addressing interoperability and privacy concerns [109]. Similarly, Steganography techniques encrypt medical records within multiple (DICOM) images using optimized Least Significant Bit (LSB) substitution, producing hidden data that are resilient against tampering and unauthorized access [110].

## 5. Future Directions

### 5.1. Prospective, Real-World Validation

The evolution of artificial intelligence in critical care is progressing along several complementary trajectories that collectively promise to transform intensive care practice. A pivotal advancement is the transition from retrospective validation toward rigorous prospective evaluation, including randomized controlled trials of AI interventions.

The TREWS sepsis study represents an early exemplar of this approach, and similar trials are now underway for various other applications [32]. These studies are crucial for establishing the real-world clinical impact of AI tools and defining their appropriate role within the broader context of critical care delivery.

Future trial designs must address the unique challenges of AI evaluation, including the following:Determining appropriate control conditions (e.g., standard care versus non-AI decision support).Handling continuous model updates during trial periods.Measuring both direct outcomes and secondary effects on workflow and team dynamics.Evaluating cost-effectiveness and resource utilization.

This level of rigorous validation is critical for distinguishing clinically meaningful AI implementations from those that merely demonstrate statistical significance without meaningful real-world clinical utility.

### 5.2. Integrated Multimodal Platforms

Concurrent with this evolution in clinical validation, the field is moving toward more sophisticated multimodal and integrated AI systems. Rather than isolated point solutions addressing single clinical problems, future AI platforms will likely integrate multiple data modalities and functions to provide comprehensive support across the critical care workflow. A fully developed ICU AI platform might simultaneously provide continuous risk monitoring for multiple complications (such as sepsis, respiratory failure, and acute kidney injury), treatment optimization recommendations based on individual patient trajectories, automated documentation and information retrieval, and support for team communication and handoffs. This integration could significantly reduce alert fragmentation and provide more contextually relevant decision support by understanding the full clinical picture rather than focusing on isolated physiological parameters. Architectural approaches that utilize foundation models as central coordination layers, with specialized modules for specific tasks, show particular promise in achieving this level of integration while maintaining performance across diverse clinical tasks [111].

### 5.3. Human–AI Co-Intelligence

As these integrated systems mature, defining optimal human–AI collaboration models becomes increasingly important. The ideal partnership between human clinicians and AI systems remains an active area of research, with different models of collaboration potentially appropriate for different clinical tasks. In certain contexts, AI may primarily serve as a screening tool that prioritizes cases for human review, enabling clinicians to focus their attention where it is most needed. In other scenarios, AI can serve as decision support, providing recommendations for human approval while augmenting clinical judgment without replacing it. For narrowly defined, low-risk tasks, AI may eventually function as an autonomous agent, handling routine aspects of care while freeing human attention for more complex cases. Perhaps most promisingly, AI could evolve as a cognitive extension that handles information processing and pattern recognition, while humans focus on complex reasoning, ethical judgment, and interpersonal care—strengths that remain uniquely human. Research on team cognition and adaptive automation will inform these evolving collaborative relationships, helping to define interaction paradigms that maximize the complementary strengths of human expertise and computational power [104].

### 5.4. AI Literacy

The increasing integration of AI into critical care necessitates a corresponding evolution in education and training for critical care professionals. Future clinicians will require competencies that extend beyond traditional medical knowledge to include a basic understanding of AI/ML principles and limitations, critical appraisal of AI-generated recommendations, recognition of situations where AI may be unreliable, and effective communication about AI inputs to patient care decisions. Medical education institutions are beginning to incorporate these topics into their training curricula, but significant work remains to define core competencies and develop effective evaluation methods [112]. This educational transformation represents not merely an addition to existing training requirements but a fundamental reconceptualization of critical care practice in an era of human–AI collaboration. Addressing these educational needs proactively will be essential to realize the potential benefits of AI while mitigating risks associated with inappropriate reliance on or rejection of computational assistance.

The convergence of these developments—rigorous clinical validation, integrated multimodal systems, refined human–AI collaboration models, and evolved educational approaches—suggests a future where AI becomes seamlessly integrated into critical care practice. This integration promises to enhance clinical decision-making, reduce cognitive burden on healthcare providers, improve resource utilization, and ultimately optimize patient outcomes. However, realizing this vision requires continued transdisciplinary collaboration among clinicians, data scientists, human factors engineers, educators, ethicists, and patients themselves. By addressing the technical, clinical, ethical, and educational dimensions of AI implementation in parallel, the critical care community can navigate toward a future where technology augments rather than replaces the human elements of care that remain essential to the discipline.

## 6. Conclusions

Artificial intelligence is poised to transform intensive care medicine through enhanced prediction, personalization, and workflow optimization. Current applications span early warning systems, sepsis management, mechanical ventilation, diagnostic support, and clinical documentation, with varying levels of evidence and implementation maturity. While significant challenges remain in technical integration, clinical workflow adaptation, explainability, and regulation, promising results from prospective studies suggest that thoughtfully designed and implemented AI tools can meaningfully improve critical care outcomes.

The future of AI in intensive care will likely involve more integrated, multimodal systems that collaborate with rather than replace human clinicians. Success will require ongoing interdisciplinary collaboration between critical care professionals, data scientists, human factor engineers, and ethicists. As the field evolves, maintaining focus on patient-centered outcomes while addressing issues of transparency, equity, and appropriate validation will be essential to realizing the full potential of AI-augmented critical care.

## Figures and Tables

**Figure 1 jcm-14-04026-f001:**
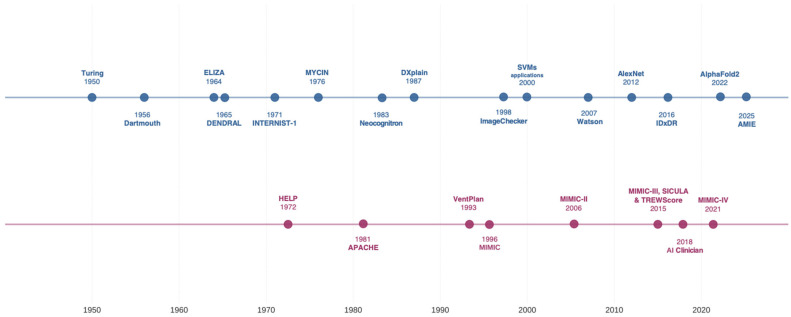
Key technological milestones in AI and ML applications for biomedicine and intensive care medicine. The upper timeline (blue) depicts landmark developments in general biomedical applications, spanning from early theoretical foundations to current specialized systems as shown in Appendix A. The lower timeline (purple) illustrates parallel innovations specific to intensive care medicine, highlighting progression from basic clinical decision support to sophisticated predictive analytics, as shown in Appendix A. A concise description of each timepoint, with corresponding scientific references, is provided in the Appendix A. Abbreviations: APACHE, Acute Physiology and Chronic Health Evaluation; CAD, computer-aided diagnosis; ELIZA, early natural language processing system; HELP, Health Evaluation through Logical Processing; MIMIC, Medical Information Mart for Intensive Care; SICULA, Super ICU Learner Algorithm; SVM, support vector machine; TREWScore, Targeted Real-time Early Warning Score.

**Figure 2 jcm-14-04026-f002:**
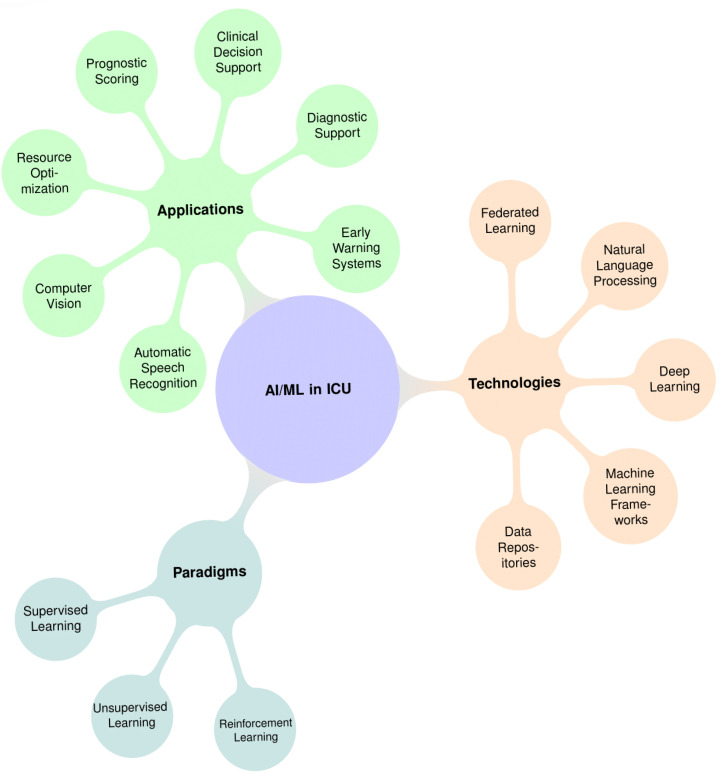
Mind-map illustrating key aspects of AI/ML integration in ICUs.

**Table 1 jcm-14-04026-t001:** Timeline of machine intelligence evolution in the biomedical field, particularly in intensive care medicine.

Year	Technology	Example	Key Innovation
1959	Logical Framework	Ledley and Lusted’s “Reasoning Foundations of Medical Diagnosis”	First formal approach to medical decision-making using symbolic logic and probability [12]
1972	Rule-based Expert System	MYCIN	First major clinical decision support system for infectious disease diagnosis and antibiotic selection [13]
1976	Bayesian Network	de Dombal’s Acute Abdominal Pain Diagnosis System	Applied Bayesian probability for differential diagnosis of abdominal pain [14,15]
1981	Causal-Associational Network	INTERNIST-I/QMR	Comprehensive internal medicine diagnostic system with disease-finding relationships [16,17]
1985	Severity Scoring System	APACHE II	First widely adopted ICU mortality prediction model using physiologic variables [18]
1993	Decision Support Framework	Arden Syntax	Standardized representation for sharing medical knowledge and decision rules [19]
1994	Integrated Clinical System	HELP System	Hospital-wide clinical decision support integrated with electronic records [20]
1997	Probabilistic Expert System	DXplain	Diagnostic decision support system using probabilistic reasoning [21]
2001	Real-time Alerting System	Medical Emergency Team Triggers	Early implementation of rule-based deterioration detection [22]
2006	Clinical Data Repository	MIMIC-II Database	First major open-access critical care database enabling ML research [23]
2010	Supervised Machine Learning	SuperLearner ICU Mortality Prediction	Ensemble ML methods outperform traditional scoring systems [24]
2014	Early Warning System	eCART	ML-based deterioration prediction with demonstrated clinical impact [25]
2016	Deep Learning	DeepPatient	Deep learning for disease prediction using EHR data [26]
2018	Reinforcement Learning	AI Clinician	RL for sepsis treatment optimization using retrospective ICU data [27,28]
2019	Unsupervised Learning	Sepsis Phenotyping	ML-identified sepsis subtypes with differential treatment responses [29,30]
2020	Temporal Deep Learning	Circulatory Failure Early Warning	Long Short-Term Memory networks for predicting hemodynamic instability from continuous data [31]
2022	Validated AI Implementation	TREWS Sepsis System	First prospectively validated AI system showing mortality reduction [32]
2023	Foundation Models	Med-PaLM/Clinical LLMs	Large language models demonstrating medical reasoning and knowledge [33]
2024	Multimodal AI Systems	Integrated Clinical AI Platforms	Systems combining multiple AI modalities for comprehensive decision support [33]

Note. APACHE II = Acute Physiology and Chronic Health Evaluation II; EHR = electronic health records; DXplain = Diagnostic Explanation System; eCART = Electronic Cardiac Arrest Risk Triage; HELP System = Health Evaluation through Logical Processing; INTERNIST-I/QMR = Quick Medical Reference for INTERNIST-I; LLM = Large language model; Med-PaLM = Medical Pathways Language Model; MIMIC = Medical Information Mart for Intensive Care; MYCIN = A Rule-Based Expert System for Infectious Diseases; TREWS = Targeted Real-time Early Warning System.

**Table 2 jcm-14-04026-t002:** Summary of AI/ML applications, modalities, and methods in intensive care.

ICU Applications	Data Modalities	AI/ML Paradigms	Methods/Architectures	Example Tools/Studies
Early warning and deterioration prediction	Time-series, tabular	Supervised learning	Random Forests, RNNs, Transformers	eCART [37], Hyland et al. [31], SICULA [24]
Sepsis prediction and triage	Tabular, Text	Supervised learning	XGBoost, ClinicalBERT	TREWS [32], Sepsis NLP Alerts [33]
ICU discharge, length of stay (LOS) prediction	Tabular, time-series	Supervised learning	Gradient Boosting, Deep MLPs	Readmission models, ICU LOS tools
Phenotyping (e.g., ARDS, sepsis)	Tabular, time-series	Unsupervised learning	K-means, LCA, Autoencoders	Seymour et al. [29], Calfee et al. [38]
Mechanical ventilation management	Time-series	Reinforcement learning	Deep Q-Networks, Policy Gradient	AI Clinician, Peine et al. [39]
Diagnostic support (imaging)	Imaging	Supervised learning (Deep Learning)	CNNs, Vision Transformers	CheXNeXt, MIMIC-CXR [40]
Clinical documentation/NLP	Text	Supervised learning, pretraining	Transformers, BERT, GPT	AutoNote, Med-PaLM
Multimodal clinical reasoning	Text + tabular + imaging	Transfer learning/fine-tuning	Foundation Models (GPT-4, Med-PaLM)	Med-PaLM, LLM-driven ICU copilots [41]

Note. ARDS = Acute Respiratory Distress Syndrome; BERT = Bidirectional Encoder Representations from Transformers; ClinicalBERT = Clinical adaptation of BERT for healthcare-specific tasks; CNNs = Convolutional Neural Networks; Deep Q-Networks = Deep Q-Learning Networks; eCART = Electronic Cardiac Arrest Risk Triage; Foundation Models = Large, generalizable AI models pre-trained on diverse datasets; GPT = Generative Pre-trained Transformer; GPT-4 = Fourth-generation version of GPT; LCA = Latent Class Analysis; LLM = Large language model; LOS = Length of stay; MIMIC-CXR = Medical Information Mart for Intensive Care Chest X-ray dataset; Med-PaLM = Medical Pathways Language Model; NLP = Natural Language Processing; Policy Gradient = Reinforcement Learning Policy Optimization Algorithm; RNNs = Recurrent Neural Networks; SICULA = Super ICU Learner Algorithm; TREWS = Targeted Real-Time Early Warning System; XGBoost = Extreme Gradient Boosting.

**Table 3 jcm-14-04026-t003:** Comparative performance, operational impact, and socio-technical considerations of human versus AI systems across key ICU domains.

ICU Domain	Human Benchmark ^‡^	AI Benchmark (Best Published)	Documented Benefit (Effect Size)	Running Cost ^†^	Key AI Advantages	Limitations and Interaction Challenges	References
Early-warning/deterioration	MEWS AUROC ≈ 0.70 (ward vital checks every 4 h)	eCART GBM AUROC ≈ 0.85; RNN model AUROC 0.90 at 6 h horizon	Δ + 0.15–0.20 AUROC; sensitivity ↑ ≈ 250% at matched specificity; median alert 2–6 h earlier	Low → Medium	Continuous streaming; multivariate pattern detection; earlier rescue team activation	High-quality signal feed; alert fatigue; model drift; opacity	Cho et al. [56]; Churpek et al. [57,58]; Hyland et al. [31]
Sepsis detection and management	Clinical recognition delay 2–6 h; guideline adherence variable	TREWS RF AUROC 0.87; alert acted ≤ 3 h dropped mortality by 3.3 pp	4–12 h earlier detection; 18.7% relative mortality ↓	Medium → High	Standardised, 24/7 screening; complex pattern handling; RL prototypes personalise fluids/pressors	Inter-site performance spread (Epic model: Se 33%, PPV 12%); data bias; causal opacity	Adams et al. [32]; Wong et al. [59]; Komorowski et al. [27]
Mechanical ventilation	Manual titration; >20% of ARDS cases plateau > 30 cmH_2_O; dyssynchrony often missed	VentAI RL policy increased protective-vent use by 203%; waveform AI detects cycling asynchrony with > 90% accuracy	Model-guided settings associated with 2–3 pp survival gain (retrospective); ↓ driving pressure	Medium → High	Patient-specific closed-loop optimisation; real-time waveform analytics; workload relief	Limited prospective safety trials; autonomy liability; cross-device generalisability	Peine et al. [39]; Gholami et al. [60]; Sottile et al. [61]
Diagnostic support (CXR ± LLM)	9 radiologists: AUROC 0.832 across 14 findings; variable fatigue	CheXNeXt * AUROC 0.846; read 420 images in 1.5 min; LLM differential top 10 accuracy of 59.1% vs. 33.6% (juniors)	Similar or better accuracy in <1% of the time; broader differential lists	High → Medium	Speed; consistency; 24/7 availability; heat-map explanations	Limited clinical context; rare/atypical cases; automation bias	Rajpurkar et al. [40]; McDuff et al. [62]
Documentation and workflow	30–50% of shift on notes; 50 min after-hours EHR time	NLP/ASR tools cut note time by 15–20%; after-hours EHR ↓ 30%; alert generation ↑ 10% speed	8% of shifts returned to bedside care; error rate ↓	Low → Medium	Administrative burden relief; real-time structured data; searchable notes	Privacy; nuance capture; language diversity; dependency on templates	Alsentzer et al. [63]; Huang et al. [64]; Archana et al. [65]
Resource optimisation	Experience-based staffing, reactive bed assignment	LOS/discharge ML models improve census forecasting; CV room activity analytics predict delirium	↑ forecast accuracy → better bed and staff utilisation (exact % context-specific)	Medium	Pro-active, data-driven planning; multi-variable optimization	Sudden exogenous shocks; fairness in allocation; staff acceptance	Bertsimas et al. [66]; Siegel et al. [67]

‡ Human benchmark denotes the best-reported clinician or rule-based performance in the comparator arms of the cited studies; AI benchmark is the highest-quality, peer-reviewed model with external or prospective validation for the same task; documented benefit is the absolute effect size taken directly from the human; * CheXNeXt, a convolutional neural network to concurrently detect the presence of 14 different pathologies in chest radiograph—AI comparison (e.g., ΔAUROC, lead-time gained, mortality, or minutes saved); † running cost categories are qualitative—Low = stand-alone software/SaaS, Medium = software plus moderate integration and compute, High = bedside hardware or continuous GPU inference with governance overhead. See the Abbreviations section for term definitions. Abbreviations: ARDS = Acute Respiratory Distress Syndrome; ASR = Automatic Speech Recognition; AUROC = Area Under the Receiver Operating Characteristic Curve; CheXNeXt = A convolutional neural network to detect 14 pathologies; CV = Computer Vision; CXR = Chest X-Ray; eCART = Electronic Cardiac Arrest Warning System; GBM = Gradient-Boosting Machine; LLMs = Large Language Models; LOS = Length of Stay; MEWS = Modified Early Warning Score; PPV = Positive Predictive Value; pp = percentage points; RF = Random Forest; RL = Reinforcement Learning; RNN = Recurrent Neural Network; Se = Sensitivity; TREWS = Targeted Real-Time Early Warning System.

**Table 4 jcm-14-04026-t004:** Implementation challenges for AI-CDSS in intensive care.

Domain	Challenge	Critical Care Relevance	Mitigation Strategies	Relevant Frameworks/Principles
Data and Infrastructure	Data fragmentation and heterogeneity	ICU systems (monitors, EHRs, devices) are poorly integrated, leading to incomplete or unsynchronized inputs	Interoperability layers; data harmonization; standard ontologies (e.g., FHIR)	ISO 80001, HL7, FAIR principles [93,94,95]
	Data quality and artifact noise	ICU data (e.g., vitals, labs) are prone to gaps, outliers, and artifacts that affect model performance	Robust preprocessing; outlier detection; real-time signal cleaning	Good Clinical Data Management Practice (GCDMP)
Model Design	Generalizability across ICUs	Models trained in one setting (e.g., tertiary ICU) often perform poorly in others due to population differences	External validation; domain adaptation; multicenter training datasets	TRIPOD-AI, MINIMAR
	Updating and model drift	Static models degrade over time; adaptive models pose safety and validation risks	Version control; locked models with scheduled retraining; monitoring performance	FDA Predetermined Change Control Plan (PCCP)
Clinical Workflow Integration	Poor timing or placement of AI support	Alerts or predictions may come too late or disrupt natural decision points	Workflow-mapped deployment; UI design aligned with ICU care pathways	Human-centered design principles
	Alert fatigue	ICU teams already face alarm overload; AI tools may add to the burden without benefit	Alert prioritization; threshold tuning; suppressive logic during emergencies	AAMI standards; ISO/IEC usability standards
	Low clinical adoption	AI-CDSS often bypassed if they are misaligned with team roles or too opaque	Co-design with ICU teams; agile iterations; training and onboarding	Implementation Science best practices
Trust and Transparency	Black box models and explainability	Clinicians must understand the rationale to use or defend AI-assisted decisions	SHAP/LIME; counterfactuals; interpretable surrogate models	EU AI Act (Transparency and Explainability clauses)
	Misalignment with clinical reasoning	AI outputs may not fit into clinicians’ diagnostic frameworks	Knowledge-augmented models; explanation interfaces using clinical logic	ACM Code of Ethics, Human–AI Teaming Guidelines
Governance and Regulation	Bias and inequity	AI may reinforce disparities based on race, sex, and comorbidities	Subgroup analysis; fairness audits; demographic-aware training	FDA GMLP; GDPR Art. 22; IEEE P7003
	Accountability and liability	Unclear who is responsible when AI recommendations contribute to harm	Maintain clinician-in-the-loop control; document AI influence in decision logs	Malpractice law (jurisdiction-specific); legal gray zones
	Privacy and data use	ICU data are highly sensitive; secondary use for training raises legal/ethical concerns	De-identification; federated learning; institutional review board (IRB) approval	HIPAA, GDPR, Institutional Ethics Committees

Note. AAMI = Association for the Advancement of Medical Instrumentation; ACM = Association for Computing Machinery; AI-CDSS = Artificial Intelligence Clinical Decision Support Systems; EHR = Electronic Health Record; EU AI Act = European Union Artificial Intelligence Act; FAIR = Findable, Accessible, Interoperable, Reusable (data principles); FDA = U.S Food and Drug Administration; FDA GMLP = U.S. FDA Good Machine Learning Practice guidelines; FHIR = Fast Healthcare Interoperability Resources; GCDMP = Good Clinical Data Management Practice; GDPR = General Data Protection Regulation; HIPAA = Health Insurance Portability and Accountability Act; HL7 = Health Level Seven International; IEEE P7003 = IEEE Standard for Algorithmic Bias Considerations; IRB = Institutional Review Board; ISO = International Organization for Standardization; IEC = International Electrotechnical Commission; LIME = Local Interpretable Model-agnostic Explanations; MINIMAR = Minimum Information for Medical AI Reporting; PCCP = Predetermined Change Control Plan; SHAP = SHapley Additive exPlanations; TRIPOD-AI = Transparent Reporting of a multivariable prediction model for Individual Prognosis Or Diagnosis—Artificial Intelligence extension; UI = User Interface.

## Data Availability

The datasets analyzed during this study are publicly available. All data supporting the conclusions of this article are included within the article and its Appendix A. The publicly available datasets analyzed are accessible through the sources cited in the references.

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
