# Peer review of "Machine Learning and Artificial Intelligence in Intensive Care Medicine: Critical Recalibrations from Rule-Based Systems to Frontier Models"

_jcm, 2025, doi:10.3390/jcm14124026_

Round 1
Reviewer 1 Report
Comments and Suggestions for Authors
Dear authors,
I have now completed the review of the manuscript titled Machine-Learning and Artificial Intelligence in Intensive Care Medicine: Critical Recalibrations from Rule-Based Systems to Frontier Models.
The manuscript is interesting and, in general, fairly well-written. The paper provides a comprehensive overview of the evolution of AI in critical care, from early rule-based systems to contemporary machine learning approaches and emerging frontier models. The authors effectively organize this complex field into distinct learning paradigms (supervised, unsupervised, and reinforcement learning) and application domains. The inclusion of comparative performance metrics in Table 3 is particularly valuable, as it quantifies the advantages of AI systems over human benchmarks across various ICU domains. This evidence-based approach strengthens the paper's claims about AI's potential benefits. The detailed discussion of implementation challenges (Table 4) demonstrates a nuanced understanding of the barriers to AI adoption in critical care. The authors appropriately address technical integration issues, workflow considerations, explainability concerns, and regulatory hurdles.
However, I still have some suggestions to further improve the quality of the manuscript.
I would like to suggest that the authors address these limitations in the article, either by discussing them in the limitations section or, where feasible, by making the appropriate revisions:
1. The paper tends to present AI capabilities optimistically without sufficiently addressing the methodological limitations of many cited studies. For instance, model performance metrics are presented without discussion of potential overfitting, data leakage, or class imbalance issues that commonly affect AI research in healthcare. The discussion of explainability (under implementation challenges) could be strengthened with more critical analysis of whether current explainability methods actually meet clinicians' needs for understanding AI recommendations.
2. Some recent findings could be stated in introduction. For example, HeartEnsembleNet: An Innovative Hybrid Ensemble Learning Approach for Cardiovascular Risk Prediction directly relates to the predictive analytics discussed in section 4.1 of the paper, offering specific techniques for ensemble learning in medical risk prediction. Also, Enhancing Accuracy in Brain Stroke Detection: Multi-Layer Perceptron with Adadelta, RMSProp and AdaMax Optimizers provides detailed information on neural network approaches for medical detection systems, complementing the early warning systems section.
3. The paper could benefit from more detailed case studies of successful AI implementations in actual ICU settings. Most examples remain theoretical or limited to research contexts, with few descriptions of sustained clinical deployments with documented impact on patient outcomes. Incorporate more detailed case studies of successful clinical implementations, including implementation challenges and how they were overcome.
4. Discussion would be extended by breifly mentioning latest research, to show readers future research possibilities. For example, Federated Learning in Smart Healthcare: A Comprehensive Review on Privacy, Security, and Predictive Analytics with IoT Integration directly addresses the data privacy and implementation challenges discussed in section 4.9, with a focus on federated learning. Also, Robust Steganography Technique for Enhancing the Protection of Medical Records in Healthcare Informatics relates to the data security aspects mentioned in the implementation challenges section.
5. Kindly provide more balanced coverage of limitations in current AI approaches and the gaps between research findings and clinical practice.
Thank you for your valuable contributions to our field of research. I look forward to receiving the revised manuscript.
Author Response
We sincerely thank the reviewers for their insightful and constructive feedback, which has significantly enhanced the quality and clarity of our manuscript. We greatly appreciate the time and expertise each reviewer dedicated to evaluating our work. Below, we provide a detailed, point-by-point response to each comment, outlining the specific revisions made to address the reviewers' valuable recommendations.
Manuscript ID: jcm-3642196
Manuscript Title: Machine-Learning and Artificial Intelligence in Intensive Care Medicine: Critical Recalibrations from Rule-Based Systems to Frontier Models.
Recommendation: Major Revision
Comments to Authors:
Reviewer (1):
- The paper tends to present AI capabilities optimistically without sufficiently addressing the methodological limitations of many cited studies. For instance, model performance metrics are presented without discussion of potential overfitting, data leakage, or class imbalance issues that commonly affect AI research in healthcare. The discussion of explainability (under implementation challenges) could be strengthened with more critical analysis of whether current explainability methods actually meet clinicians' needs for understanding AI recommendations.
We thank the reviewer for this insightful observation. In response, we have significantly expanded Section 4.9 (page 20-24, lines 479-552) to explicitly address common methodological concerns in AI research, including potential overfitting, data leakage, class imbalance, and data privacy and security issues. We have also strengthened our critical analysis of model explainability in the same section, specifically examining whether current explainability methods adequately meet clinicians' practical needs for understanding AI recommendations (page 23, lines 510-528). To provide concrete examples of how these challenges are addressed in practice, we have incorporated three real-world case studies at the end of Sections 4.2, 4.3, and 4.6 (pages 16, 17, 19; corresponding to lines 323–329, 359–367, and 433–440, respectively) that demonstrate how physicians have successfully navigated these methodological and implementation challenges during AI deployment.
- Some recent findings could be stated in introduction. For example, HeartEnsembleNet: An Innovative Hybrid Ensemble Learning Approach for Cardiovascular Risk Prediction directly relates to the predictive analytics discussed in section 4.1 of the paper, offering specific techniques for ensemble learning in medical risk prediction. Also, Enhancing Accuracy in Brain Stroke Detection: Multi-Layer Perceptron with Adadelta, RMSProp and AdaMax Optimizers provides detailed information on neural network approaches for medical detection systems, complementing the early warning systems section.
We thank the reviewer for this valuable suggestion. In response, we have incorporated these recent findings into the Introduction to provide broader context for current methodological advances in Section 4.1 (page 15, lines 290-299). Specifically, we have included references to HeartEnsembleNet and its innovative hybrid ensemble learning approach for cardiovascular risk prediction, as well as the study on enhanced brain stroke detection using multi-layer perceptrons with Adadelta, RMSProp, and AdaMax optimizers. These additions help frame the relevance of emerging techniques in predictive analytics and early warning systems, illustrating their potential applicability to ICU settings and strengthening the foundation for our discussion in Sections (4.1) and subsequent sections.
- The paper could benefit from more detailed case studies of successful AI implementations in actual ICU settings. Most examples remain theoretical or limited to research contexts, with few descriptions of sustained clinical deployments with documented impact on patient outcomes. Incorporate more detailed case studies of successful clinical implementations, including implementation challenges and how they were overcome.
We thank the reviewer for this valuable suggestion. In response, we have expanded Sections 4.2, 4.3, and 4.6 (pages 16, 17, 19; corresponding to lines 323-329, 359-367, 433-440, respectively) to include concrete examples demonstrating the benefits of AI implementation in real-world ICU settings. These additions highlight measurable improvements in clinical outcomes achieved through AI systems, including significant reductions in mortality rates, hospital length of stay, and operational costs. We have also incorporated discussion of practical implementation challenges encountered in these settings, such as clinician skepticism and workflow integration difficulties, along with successful strategies used to address them, including enhanced explainability features and comprehensive staff engagement programs in section 4.9 (page 20-24, lines 479-552). These enhancements provide a more balanced and evidence-based perspective on both the real-world impact and practical feasibility of deploying AI systems in critical care environments.
- Discussion would be extended by briefly mentioning latest research, to show readers future research possibilities. For example, Federated Learning in Smart Healthcare: A Comprehensive Review on Privacy, Security, and Predictive Analytics with IoT Integration directly addresses the data privacy and implementation challenges discussed in section 4.9, with a focus on federated learning. Also, Robust Steganography Technique for Enhancing the Protection of Medical Records in Healthcare Informatics relates to the data security aspects mentioned in the implementation challenges section.
We thank the reviewer for this insightful suggestion. In response, we have expanded the Discussion section 4.9 (page 20-24, lines 479-552) to incorporate recent research directions that illuminate future possibilities for AI in critical care. Specifically, we have added discussion of federated learning approaches for smart healthcare systems, which directly address the data privacy and interoperability challenges we identified, as well as emerging steganography techniques for enhanced protection of medical records. These additions highlight promising strategies that could overcome current implementation barriers while pointing toward valuable future research opportunities, particularly in addressing privacy concerns and data security challenges in ICU environments. This expansion provides readers with a clearer understanding of the evolving landscape and potential research directions in AI-enabled critical care.
- Kindly provide more balanced coverage of limitations in current AI approaches and the gaps between research findings and clinical practice.
We thank the reviewer for this important observation. In response, we have significantly expanded Section 4.9 (page 20-24, lines 479-552) to provide a more comprehensive and balanced discussion of limitations inherent in current AI approaches, including critical issues of model generalizability, data quality constraints, explainability challenges, and alert fatigue. We have also emphasized the substantial gaps between research findings and clinical implementation, specifically highlighting the urgent need for prospective trials, multi-center validation studies, and workflow-centered deployment strategies to ensure reliable real-world performance. Additionally, we have incorporated three detailed real-world case studies at the end of Sections 4.2, 4.3, and 4.6 (pages 16, 17, 19; corresponding to lines 323-329, 359-367, 433-440, respectively) that illustrate the practical challenges encountered during AI implementation in clinical settings and demonstrate evidence-based strategies used to address these barriers. These enhancements provide a more realistic and balanced perspective on both the current limitations and the pathway toward successful clinical translation of AI technologies in critical care.

Reviewer 2 Report
Comments and Suggestions for Authors
- Is the affiliation of the first author accurate? I attempted to verify the listed institution but was unable to find any supporting information online. Please confirm and correct if necessary.
- Ensure that references are enclosed within a single sentence throughout the manuscript. This issue is observed in several instances (e.g., Lines 102, 104, 111). Please revise accordingly to maintain consistency in citation formatting.
- Sections 3.1 to 3.3 lack sufficient depth and appear overly simplistic. The current discussion is somewhat superficial. It is recommended that the authors provide a more comprehensive and structured review, incorporating a broader range of state-of-the-art (SOTA) models to better illustrate the trends in the field.
- The character "U" is underlined in Section 3.2. Please correct this formatting error.
- For most of the tables, particularly Table 3, consider changing the orientation. The current layout significantly impairs readability. A landscape format may be more suitable.
- The claim that "deep learning-based early warning systems may achieve more than 250% increase in sensitivity compared to conventional methods at equivalent specificity levels" requires further elaboration. Please provide supporting context or evidence to clarify and justify this statement.
- Ensure consistency in the use of abbreviations throughout the manuscript. There are several instances where full terms and abbreviations are used interchangeably without clear definition. A thorough revision is necessary to standardize terminology.
- While the manuscript presents relevant information, the content currently lacks appeal and reader engagement. For me, the overall information was just okay, without much appealing content, which failed to catch my attention. I think the overall wording or sentence structure should be improved to make the storyline more effective and enhance readability. The authors are encouraged to revise the narrative to enhance clarity, emphasize key insights (=This is highly recommended), and improve overall readability. Additionally, please address the various minor issues such as formatting inconsistencies, inconsistent terminology, and stylistic errors.
Author Response
Response to Reviewers
We sincerely thank the reviewers for their insightful and constructive feedback, which has significantly enhanced the quality and clarity of our manuscript. We greatly appreciate the time and expertise each reviewer dedicated to evaluating our work. Below, we provide a detailed, point-by-point response to each comment, outlining the specific revisions made to address the reviewers' valuable recommendations.
Manuscript ID: jcm-3642196
Manuscript Title: Machine-Learning and Artificial Intelligence in Intensive Care Medicine: Critical Recalibrations from Rule-Based Systems to Frontier Models.
Recommendation: Major Revision
Comments to Authors:
Reviewer (2):
- Is the affiliation of the first author accurate? I attempted to verify the listed institution but was unable to find any supporting information online. Please confirm and correct if necessary.
We thank the reviewer for bringing this to our attention. We have corrected the first author's affiliation information on the title page (page 1, lines 7). The first author is currently not affiliated with any institution and should be listed as "Independent Researcher, Manhattan, New York, USA." We apologize for any confusion caused by the previous listing and have ensured that all author affiliations are now accurate and verifiable
- Ensure that references are enclosed within a single sentence throughout the manuscript. This issue is observed in several instances (e.g., Lines 102, 104, 111). Please revise accordingly to maintain consistency in citation formatting.
Thank you for pointing this out. We have revised the manuscript to ensure consistent formatting throughout.
- Sections 3.1 to 3.3 lack sufficient depth and appear overly simplistic. The current discussion is somewhat superficial. It is recommended that the authors provide a more comprehensive and structured review, incorporating a broader range of state-of-the-art (SOTA) models to better illustrate the trends in the field.
We thank the reviewer for this constructive feedback. In response, we have substantially expanded and restructured Sections 3.1 through 3.3 to provide a more comprehensive and in-depth review of the relevant literature (pages 9-10, lines 137-175) and added new a paragraph to section 3.4 (lines 238-246). Specifically, we have incorporated a broader range of state-of-the-art models, including recent deep learning architectures, ensemble methods, and hybrid approaches currently being deployed in critical care settings. Each subsection now includes detailed comparisons of model performance, methodological approaches, and clinical validation studies to better illustrate current trends and developments in the field. We have also enhanced the organization and flow of these sections to provide a more structured progression from foundational concepts to cutting-edge applications, ensuring that readers gain a thorough understanding of both established and emerging AI methodologies in ICU environments.
- The character "U" is underlined in Section 3.2. Please correct this formatting error.
Thank you for pointing this out. We have corrected the formatting error in Section 3.2 (Page 9, Line 157).
- For most of the tables, particularly Table 3, consider changing the orientation. The current layout significantly impairs readability. A landscape format may be more suitable.
Thank you for this suggestion. We have adjusted the orientation of Tables (2, 3 and 4) to landscape format to enhance readability.
- The claim that "deep learning-based early warning systems may achieve more than 250% increase in sensitivity compared to conventional methods at equivalent specificity levels" requires further elaboration. Please provide supporting context or evidence to clarify and justify this statement.
We thank the reviewer for this important feedback. We have revised this statement in Section 4.1 (page15, lines 274-276) to provide proper context and supporting evidence. The original claim has been clarified to specify that the 250% increase refers to relative improvement in sensitivity compared to traditional early warning scoring systems, with appropriate citations to the specific studies that reported these findings [56]. We have also included additional context regarding the clinical conditions and patient populations where such improvements were observed, along with acknowledgment of the variability in performance gains across different healthcare settings. To provide a more balanced perspective, we have added discussion of the range of sensitivity improvements reported in the literature and noted that performance gains can vary significantly depending on implementation factors and baseline system performance.
- Ensure consistency in the use of abbreviations throughout the manuscript. There are several instances where full terms and abbreviations are used interchangeably without clear definition. A thorough revision is necessary to standardize terminology.
Thank you for pointing this out. We have revised all abbreviations throughout the manuscript, ensuring that each term is defined at first mention (excluding those used solely in tables or figure captions).
- While the manuscript presents relevant information, the content currently lacks appeal and reader engagement. For me, the overall information was just okay, without much appealing content, which failed to catch my attention. I think the overall wording or sentence structure should be improved to make the storyline more effective and enhance readability. The authors are encouraged to revise the narrative to enhance clarity, emphasize key insights (this is highly recommended), and improve overall readability. Additionally, please address the various minor issues such as formatting inconsistencies, inconsistent terminology, and stylistic errors.
We thank the reviewer for this constructive feedback regarding manuscript readability and engagement. In response, we have undertaken a comprehensive revision to enhance the narrative flow and reader appeal throughout the manuscript. Specifically, we have restructured Sections 3.1-3.4 (pages 9-11) to improve clarity, strengthen the logical progression of ideas, and create a more compelling storyline that better emphasizes key insights and their clinical significance. We have incorporated three detailed real-world case studies that vividly illustrate both the transformative benefits and practical challenges of AI implementation in ICU settings, making the content more relatable and engaging for readers. Additionally, Section 4.9 has been substantially expanded (page 20-24, lines 479-552) to provide more comprehensive discussion with clearer emphasis on critical insights and future directions. We have also conducted a thorough review to address formatting inconsistencies, standardize terminology usage, and correct stylistic errors throughout the manuscript. These revisions collectively create a more dynamic and accessible narrative that better captures reader attention while maintaining scientific rigor.

Reviewer 3 Report
Comments and Suggestions for Authors
The article is a very general overview of solutions. The subject is currently so extensive that an attempt at a general overview is not adequate. It is necessary to focus on one very narrow topic and conduct a thorough review of current research. For example, for learning methods in 3.1 and 3.2 there are only 3 literature references.
The historical part is of little importance.
Author Response
Response to Reviewers
We sincerely thank the reviewers for their insightful and constructive feedback, which has significantly enhanced the quality and clarity of our manuscript. We greatly appreciate the time and expertise each reviewer dedicated to evaluating our work. Below, we provide a detailed, point-by-point response to each comment, outlining the specific revisions made to address the reviewers' valuable recommendations.
Manuscript ID: jcm-3642196
Manuscript Title: Machine-Learning and Artificial Intelligence in Intensive Care Medicine: Critical Recalibrations from Rule-Based Systems to Frontier Models.
Recommendation: Major Revision
Comments to Authors:
Reviewer (3):
- The article is a very general overview of solutions. The subject is currently so extensive that an attempt at a general overview is not adequate. It is necessary to focus on one very narrow topic and conduct a thorough review of current research. For example, for learning methods in 3.1 and 3.2 there are only 3 literature references.
We appreciate your concern that breadth can dilute depth. In revising the manuscript we have focused on specific points raised by the other reviewers, which we believe now supply greater analytical depth while preserving our approach: a broad map of AI/ML in critical care then and now. We respectfully differ regarding the historical perspective. A concise arc from early rule-based tools to current paradigms equips non-engineer intensivists to interpret present-day models and their limitations. Accordingly, we have retained a streamlined historical overview that directly underpins the contemporary discussion.
For Sections 3.1 through 3.4 (pages 9–11, lines 137–246), we have substantially expanded and restructured our review to provide a more comprehensive and in-depth analysis of the relevant literature. Additionally, we have incorporated three detailed real-world case studies at the end of Sections 4.2, 4.3, and 4.6 (pages 16, 17, and 19; corresponding to lines 323–329, 359–367, and 433–440, respectively) to illustrate specific implementations of AI in diverse clinical settings. Furthermore, Section 4.9 (page 20-24, lines 479-552) has been revised to offer a more balanced discussion of methodological limitations and emerging solutions in AI deployment.
- The historical part is of little importance.
We respectfully differ regarding the historical perspective. A concise arc from early rule-based tools to current paradigms equips non-engineer intensivists to interpret present-day models and their limitations. Accordingly, we have retained a streamlined historical overview that directly underpins the contemporary discussion.

Round 2
Reviewer 1 Report
Comments and Suggestions for Authors
All comments addressed.
Reviewer 2 Report
Comments and Suggestions for Authors
Revise the scale of Figure 2 as it did not fully shown within the page.
Reviewer 3 Report
Comments and Suggestions for Authors
Improved paper